# Bidirectional genetic links between chronic obstructive pulmonary disease and frailty: Genome-wide association study insights

**Fuhui Yan**[1], **Tong Wu**[2], **Qiang Meng**[3], **Feng Qu**[3]*

1 College of Clinical Medicine, Jining Medical University, Jining, Shandong, China, 2 School of Pharmacy, The Affiliated Hospital of Hangzhou Normal University, Hangzhou Normal University, Hangzhou, P. R. China, 3 Affiliated Jining First People's Hospital of Shandong First Medical University, Jining, Shandong, China

* rmyyzzeq@163.com

## Abstract

### Background

Recent research underscores a potential correlation between chronic obstructive pulmonary disease (COPD) and frailty, suggesting a shared genetic foundation. However, specific genetic factors and mechanisms underlying this association remain unclear. This study aimed to explore genetic connections between COPD and frailty using genome-wide association studies to enhance our understanding and improve clinical management and prevention strategies for these conditions.

### Method

We utilised summary statistics for genome-wide association studies to examine the genetic correlations between COPD and frailty using linkage disequilibrium score regression. Local genetic correlations were evaluated using the ρ-heritability estimates from summary statistics method. Using the established two-sample Mendelian randomization approach, causal relationships have been identified. Shared genetic variants were quantified using a bivariate causal mixture model. Shared loci and single nucleotide polymorphisms were identified by conjoint false discovery rate (conjFDR). Gene enrichment and transcriptome-wide association studies (TWAS) were conducted to explore potential transcriptomic associations across tissues.

### Results

We observed a significant genetic correlation between COPD and frailty ($Rg = 0.4324$, $P = 6.09 \times 10^{-26}$). MiXeR estimated 3,200-shared causal variants. Additionally, we discovered 16 shared loci linked to 91 genes, offering novel insights into gene expression across diverse tissues. The TWAS revealed 25 shared genes, representing a significant advance in understanding the genetic overlap between COPD and frailty. Furthermore, out of the 25 SNPs identified through TWAS, 4 overlapped with the lead SNPs, specifically [HLA-DRB1, PBX3, SLC22A5/OCTN2, SLMAP].

**Data availability statement:** GWAS summary statistics for COPD (ID: finngen_R10_J10_COPD) and frailty (ID: ebi-a-GCST90020053) are available from the NHGRI-EBI GWAS Catalog (https://www.ebi.ac.uk/gwas) and FinnGen database (https://www.finngen.fi/en). Access to UK Biobank data requires an application via https://www.ukbiobank.ac.uk.

**Funding:** The author(s) received no specific funding for this work.

**Competing interests:** The authors have declared that no competing interests exist.

## Conclusions

Our study shows a common genetic foundation for COPD and frailty, identifying multiple shared loci and offering insights into their underlying causal connections. These findings enhance our understanding of the biological mechanisms linking these conditions and may guide future research and treatment strategies for related diseases.

## Introduction

Chronic obstructive pulmonary disease (COPD), now the third leading cause of global mortality [1], imposes a significant economic burden. In 2017, chronic respiratory diseases afflicted 549 million individuals, with COPD constituting approximately 55% of these cases [2]. This increase in COPD in recent decades is primarily attributed to population aging and prolonged risk factor exposure, and this trend is expected to persist [3]. Frailty, a prevalent geriatric syndrome, is characterised by diminished physiological reserves, increased susceptibility to injury, and decreased stressor resistance, potentially leading to clinical events triggered by minor external stimuli, with a prevalence among the elderly ranging from 11 to 60.6% [4,5]. Frailty occurrence is closely linked to factors such as inflammation, nutrition, genetics, and endocrine influences [6] involving multiple organ system damage. The co-occurrence of COPD and frailty has gained significant attention lately, with mounting evidence indicating a link between the two [7,8]. Patients with COPD face twice the risk of frailty as their age-matched counterparts without it [9]. Frailty affects chronic lung diseases, and, reciprocally, chronic lung diseases increase frailty risk. Patients with chronic lung conditions demonstrate a greater predisposition to frailty, demonstrating frailty signs years earlier than healthy community-dwelling populations [8]. Certain biological features of chronic lung diseases reflect potential frailty-associated mechanisms, including chronic systemic inflammation, malnutrition, decreased activity, fatigue, muscle weakness, and slower gait speed (with the latter three being phenotypic criteria of frailty) [8], suggesting that these conditions share common pathophysiological roots with pathological, epidemiological, and clinical correlations [8,10] While significant research has highlighted the connections between COPD and frailty, the biological mechanisms underlying their potential comorbid relationship remain elusive.

Twin studies indicate a potential genetic influence on increased frailty risk. Heritability estimates for the syndromic and cumulative deficit frailty index are 25% and 30%, respectively, revealing significant genetic (0.57) and environmental (0.44) correlations between the indices. Monozygotic twins exhibit higher intragroup correlations for both frailty measures than dizygotic twins [11]. Furthermore, 45% of the interindividual variability in the frailty index is genetic, and 52% is attributed to environmental factors. Similarly, consistent evidence has emphasised a genetic influence on COPD onset [12]. Multiple genome-wide association studies (GWAS) have revealed an increasing number of single nucleotide polymorphisms (SNPs) and susceptibility loci as causative genes for COPD and frailty [12–14]. COPD and frailty have been demonstrated as polygenic diseases, wherein their genetic foundations are influenced by several common variants with subtle effects alongside a few rare variants that exert pronounced effects.

Existing randomised controlled trials (RCTs) and numerous cross-sectional studies have elucidated the correlations between COPD and frailty. However, the genetic causality or potential mechanistic links remain unclear, warranting further investigation. 6 Traditional RCTs, although regarded as the gold standard for directly establishing causality, encounter challenges in uncovering the shared genetic basis of common pathophysiologies and complex comorbidities of the conditions. Mendelian randomisation (MR), a method that uses genetic

variants as instruments to identify causal relationships between risk factors and disease outcomes, has been employed to uncover the causal link between frailty and COPD. However, MR has certain limitations, such as susceptibility to pleiotropy, potential confounding, and weak instrument bias. Moreover, the potential shared pathological mechanisms between frailty and COPD deserve further attention and investigation. Quantile-quantile (Q-Q) plots—a graphical tool—enable the evaluation of GWAS data quality and reliability of genetic research findings through visualisation of the genetic overlap between two traits. The conjoint false discovery rate (conjFDR) method enhances the identification of shared genetic loci by integrating data from multiple traits, thus reducing false positives and improving robustness.

Furthermore, molecular genetic research methods employed in exploring mechanisms linking COPD and frailty are lacking. Today, leveraging summary statistics from extensive GWAS facilitates examination of the genetic basis of their observed phenotypic association to a certain extent. Employing matched models and algorithms facilitates intuitive prediction of their causal relationship and investigation into their potential shared genetic basis.

This research aims to uncover the genetic links between chronic obstructive pulmonary disease (COPD) and frailty by integrating various advanced algorithms and models, and to elucidate their shared genetic mechanisms. Utilizing the conjFDR method, we identified and confirmed 16 loci with pleiotropic effects across these traits, including 7 newly discovered ones. This advancement will aid future efforts in screening and treating comorbid conditions. Through functional annotation and gene set enrichment analyses, we delineated molecular pathways and human phenotypes related to these loci. Overall, these discoveries enhance our comprehension of the genetic framework of COPD and frailty, offering new perspectives on their comorbid mechanisms and identifying potential susceptibility factors for the frailty-COPD comorbidity.

## Methods

### Study populations

We integrated datasets from European ancestry individuals, drawing from sources including FinnGen (https://www.finngen.fi/en) [15], UK Biobank, and TwinGene. Data on COPD was collected from the Finn databa, and GWAS summary statistics for COPD (ID: finngen_R10_ J10_COPD), frailty (ID: ebi-a- GCST90020053). Characterisation of COPD and frailty were conducted using ICD classification codes, with FinnGen contributors employed to refine the analysis, considering variables such as age, sex, principal components, and batch effects through a mixed-model logistic regression approach [15], The frailty index (FI) was derived from updated GWAS summary data, incorporating contributions from European individuals in the UK Biobank and TwinGene [16], This index encompassed 49 and 44 self-reported items comprising symptoms, disabilities, and diagnosed diseases from the UK Biobank and TwinGene, respectively. The FI score ranged from 0–27 [16]. Standardisation techniques were applied to the GWAS summary datasets, mitigating biases introduced via quality control processes. The analysis excluded indels and rare and low-frequency variants with a minor allele frequency of < 1% and focused solely on autosomal chromosomes.

### Study design

**Linkage disequilibrium score regression (LDSC) analysis.** Details of these methods are provided in Fig 1. Following the GWAS, we conducted a post-GWAS analysis of correlations between COPD and frailty. LDSC analysis provided a genetic correlation estimate (Rg from −1 to 1) for the two traits, reflecting their true causal effects. A genetic correlation score of +1 indicates a complete positive genetic association between the two traits, where shared

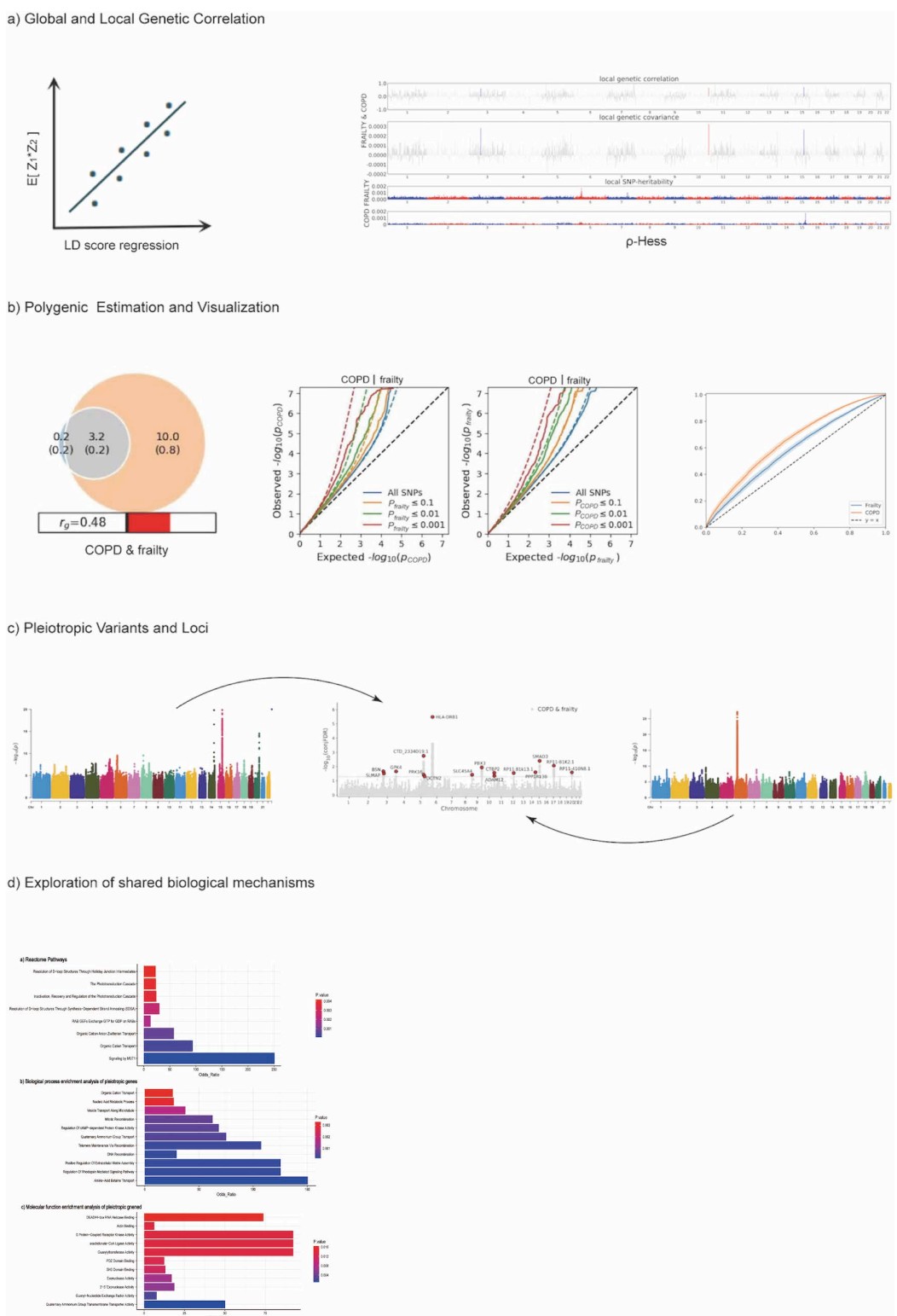

**Fig 1. Study Workflow.** We conducted a comprehensive pleiotropy analysis of COPD and frailty. GWAS, genome-wide association study; LD, linkage disequilibrium; ρ-HESS, heritability estimation from summary statistics.

genetic variants influence the traits in the same direction. Conversely, a score of –1 denotes a complete negative genetic association, where shared genetic variants exhibit opposite effects on the two traits [17]. Participants of European descent were the focus of the LDSC analysis to mitigate potential biases from differences in population genetic structures. They were included in further analyses of each trait that correlated with COPD.

## Local genetic correlation

To identify genomic regions disproportionately contributing to the global genetic correlation between COPD and frailty, we employed heritability estimates from summary statistics (ρ-HESS) to examine the local heritability of each condition and their genetic correlations within independent linkage disequilibrium (LD) blocks.

These LD blocks were obtained from the 1kG reference panel in three steps: preparing LD blocks and eigenvalues, estimating local SNP heritability for each trait, and determining local genetic covariance and standard error [18]. HESS precisely quantifies the covariance of each SNP in the genome using linear mixed models to clarify local region heritability. This approach showed the local genetic correlation between COPD and frailty, offering insights into the shared genetic basis of the diseases. All GWAS data were limited to individuals of European descent, and they were adjusted for multiple testing using Bonferroni correction based on the original method (two-tailed $P < 0.05/1703$) [19].

## Shared causal variants estimation and identification

We employed the statistical tool MiXeR (v1.3) to quantify polygenic overlap via a bivariate causal mixture model, using GWAS summary data to identify genetic factors shared between COPD and frailty. Initially, Z-scores were computed using sumstats.py, as previously described [20]. Subsequently, a bivariate cross-trait analysis was conducted to estimate shared and unique genetic variants. MiXeR results were illustrated in a Venn diagram, displaying distinct and shared polygenic overlaps between COPD and frailty. Conditional Q-Q plots were generated using all SNPs from the primary trait and its subsets at $P \leq 0.1$, $P \leq 0.01$, and $P \leq 0.001$, comparing observed values against expected $-\log_{10}$ P-values, allowing visualisation cross-trait enrichment. This enrichment, depicted in the conditional Q-Q plots as a continuous leftward shift from the null distribution, can be directly interpreted using the true discovery rate $(1 - FDR)$ [21,22].

We employed the conjoint false discovery rate (conjFDR) statistical framework to identify genetic loci shared between frailty and COPD, facilitating simultaneous discovery of SNPs significantly associated with both traits. ConjFDR—an expansion of condFDR—recalibrates GWAS test statistics for the primary phenotype using additional genetic information, thus enhancing the accuracy of genetic locus identification [22–24].

## Functional annotations of genomic risk loci and GTEx tissue enrichment analysis

Loci shared between COPD and frailty were identified using FUMA for gene mapping. We isolated significant SNPs with a conjunction of FDR < 0.05 and an independent threshold of r2 < 0.60. A refined set of lead SNPs was then curated, selecting only those remaining significant and exhibiting linkage disequilibrium at an r2 < 0.10. LD patterns were annotated using the European ancestry reference panel from the 1000 Genomes Project. Subsequently, we examined the directional effects of alleles within SNPs identified via conjFDR. This involved comparing the z-scores from the initial COPD GWAS with those from the frailty GWAS. We then assessed the novelty of each pleiotropy locus using the GWAS catalogue and

Phenoscanner database. Loci correlated with relevant phenotypes in previous studies at the genome-wide significance level (P-value = $5e^{-8}$) were considered previously identified shared loci. However, those lacking significant correlations were deemed novel. Implicated genes shared between COPD and frailty (located outside the LD major histocompatibility complex [MHC] regions) were pooled for enrichment analysis. The resulting gene list underwent Enrichr analysis for gene set enrichment, with corrections for multiple comparisons using the Benjamini-Hochberg method. Leveraging functional mapping and annotation (enrichment) [25], we conducted a GTEx tissue enrichment analysis on 35 tissue types using GTEx version 8 [26].

## Transcriptome-wide association studies

We conducted transcriptome-wide association studies (TWAS) using FUSION software to investigate the gene-expression link between COPD and frailty across specific tissues [27]. This involved integrating GWAS and eQTL summary statistics within the TWAS framework to assess the relationship between gene-expression patterns and traits. We selected trait-relevant tissue gene-expression data from the GTEx v8 project as a reference. Individual TWAS analyses for each tissue and trait combination, along with cross-analyses, facilitated identification of shared gene-tissue associations across traits. To mitigate false positives, we applied the Bonferroni correction method for multiple comparisons across all tissues [26].

## Results

### Genetic correlation

We employed cross-trait LDSC to evaluate the genetic correlation between COPD and frailty, and the results indicated a significant positive genetic correlation between them ($R_g$ = 0.43; P = $6.09 \times 10^{-26}$).

### Local genetic correlation

Using ρ-HESS, a local genetic correlation analysis of COPD and frailty revealed 167 loci with correlations (P < 0.05). Three genomic areas specifically showed significant local genetic correlations (Fig 2), indicating potential shared genetic predispositions for these conditions: 3q39 (chromosome 3:56433907-58157519, P = 0.0342), 10q78 (chromosome

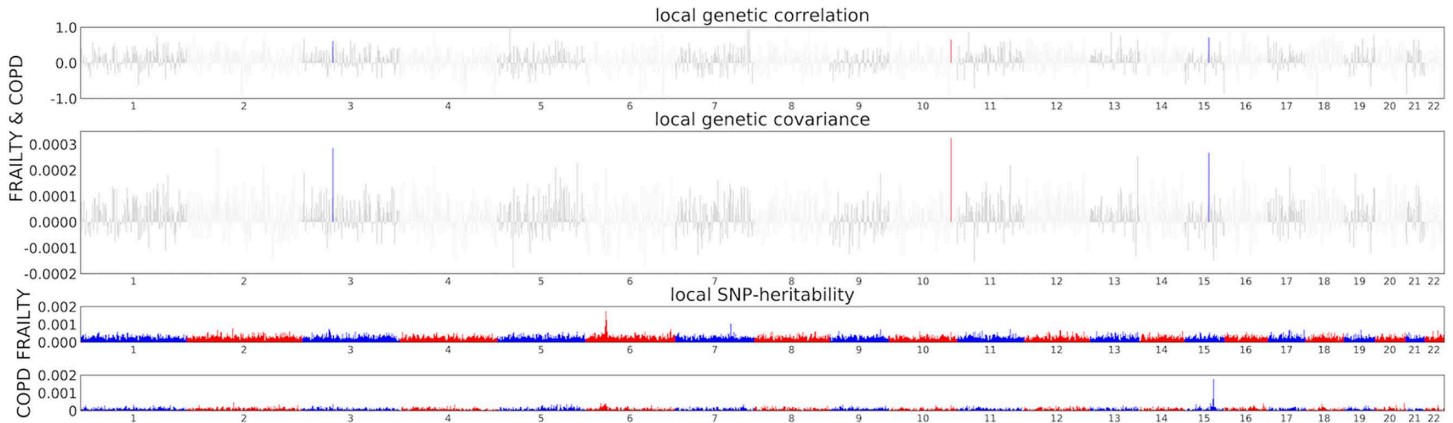

**Fig 2. Local genetic covariance estim.**

10:125869346-128001098, P = 0.0132), and 15q31 regions (chromosome 15:67094767-69017999, P = 0.00606). Even after adjusting for multiple tests with Bonferroni correction (P < 0.05/1703 = 2.93 × 10⁻⁵), the significance persisted (Supplementary table 1 in S1 File).

## Genetic correlation and polygenic overlap

Using GWAS summary statistics, MiXeR revealed polygenic overlap between COPD and frailty, indicating shared causal variants [28]. A Venn diagram (Fig 3a) illustrates that the estimated number of casual variants shared between COPD and frailty was 3.2k (SD = 0.2 k), with 0.2 k variants affecting COPD and 10.0 k (0.8 k) affecting frailty. The Dice coefficient (mean) for variants shared between COPD and frailty was 0.38. MiXeR was employed to calculate the genetic correlation among shared variants at $\rho\beta$ = 0.9986 (SD = 0.019) and the

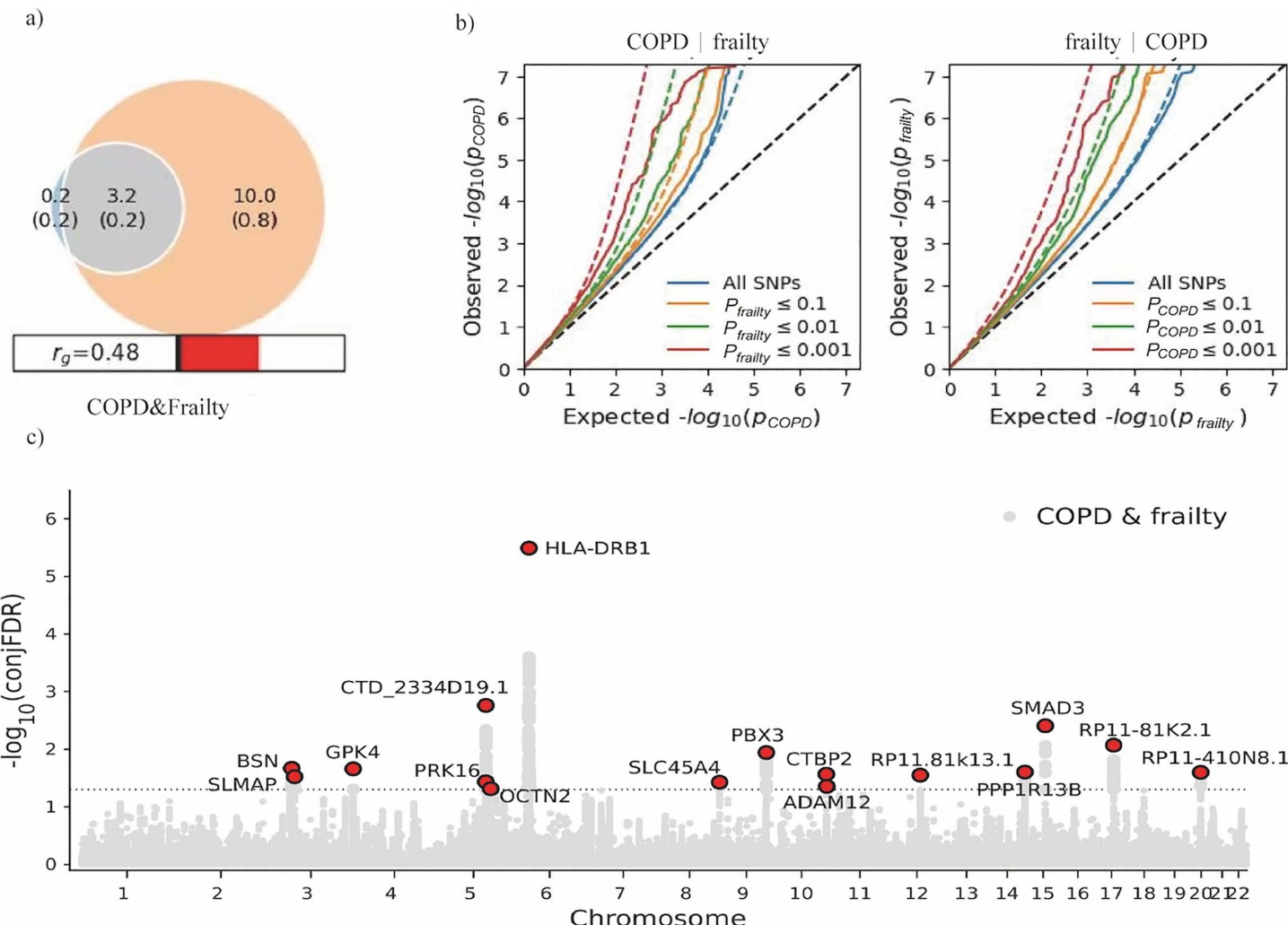

**Fig 3. Polygenic overlap between COPD and frailty.** a) The estimated number of causal variants shared (grey) between COPD and frailty calculated in MiXeR, and the genetic correlation estimated by LDSC. b) Conditional Q-Q plots of nominal versus empirical −log10 transformed P values in the primary phenotype as a function of significance of SNP associations with the secondary phenotype at the level of P ≤ 1.00 (all SNPs, blue lines), P ≤ 0.1 (red lines), P ≤ 0.01 (yellow lines) and P ≤ 0.001 (purple lines). The dashed line is the expected Q-Q plot under the null hypothesis. c): Common genetic variations associated with COPD and frailty have a conjoint false discovery rate (conjFDR) of <0.05. On the Manhattan plot, each SNP −log₁₀ transformed conjFDR value is shown on the y-axis, with chromosomal positions displayed on the x-axis. A horizontal dashed line represents the threshold for significant shared associations.

proportion of shared causal variants consistently affecting COPD and frailty at 98.73% (SD = 0.01),. This suggests a substantial proportion of the genetic foundation of COPD and frailty is shared, with shared genetic variants consistently affecting both conditions in most cases. Conditional Q-Q plots were generated to visualise the pleiotropic enrichment of SNP associations between COPD and frailty [22]. The curves for COPD-related frailty and frailty-related COPD significantly shifted leftward, indicating a significant enrichment of variants from one trait in the other. Furthermore, SNPs with higher significance for the conditional trait exhibited more pronounced leftward shifts in both plots. Bivariate analyses and Q-Q plots confirmed a substantial genetic overlap between COPD and frailty (Supplementary table 2 in S1 File).

## Shared causal variant estimation and identification

Using conjFDR, we detected genetic loci, genome wide, that were shared between COPD and frailty. After pruning, 16 distinct lead SNPs (r2 < 0.2) were identified (Fig 3c) with conjFDR < 0.05 that correlated with both COPD and frailty. All were nominally significant (P < 0.05). Among these, 11 were not discovered in the original COPD GWAS, eight were not identified in the original frailty GWAS, and seven were entirely new lead SNPs not previously associated with COPD or frailty. The z-score computation for shared loci revealed consistent effects, where all SNPs showed the same direction of correlation between COPD and frailty (i.e., one allele predicted a higher COPD risk and increased frailty). Furthermore, we conducted a detailed examination of 16 loci shared between COPD and frailty. Among them, rs9270664 exhibited the strongest shared signal overall ($P_{FUMA}$ = 3.23 × $10^{-6}$), situated in the intergenic region of 6p21.32, followed by SNP rs12523352 ($P_{FUMA}$ = 1.73 × $10^{-4}$) and rs12441344 ($P_{FUMA}$ = 3.91 × $10^{-4}$) as the second and third, respectively.

Among these 16 lead SNPs, marked pleiotropic loci were identified, including a new lead SNP, rs12441344 ($P_{FUMA}$ = 3.91 × $10^{-4}$), located at 15q22.33 near SMAD3. Studies have increasingly emphasised the importance of the TGF-β/SMAD3 signalling pathway in COPD pathology [29]. Furthermore, GDF11—a factor in frailty—influences frailty by inducing muscle atrophy through the SMAD2 signalling pathway [30]. ADAM12, a disintegrin and metalloproteinase typically associated with skeletal muscle development and regeneration, has been found to mitigate muscle degeneration and inflammatory responses in patients with muscular dystrophy [31] (Supplementary table 3 in S1 File).

## Functional genomic risk loci annotations

To visualise the genomic distribution of shared variants and their closest genes, we generated a "conjFDR Manhattan plot" for loci shared between COPD and frailty. This plot displayed all unpruned SNPs, with independently significant lead SNPs highlighted in black (Fig 3c). Additionally, using FUMA [30], we linked 522 candidate SNPs, encompassing 16 shared independent loci, with 91 protein-coding genes.

Using GTEx data, we identified genes with consistent effect directions in COPD and frailty that were significantly overexpressed in the colon. At loci with consistent effect directions, GO analysis of these genes showed four significantly associated biological processes, including 'Amino-Acid Betaine Transport', 'Telomere Maintenance Via Recombination', 'Inositol Phosphate Biosynthetic Process', and 'Quaternary Ammonium Group Transport'. Additionally, two cellular component gene sets were significantly related: 'Brush Border Membrane' and 'Heterotrimeric G-protein Complex'. Moreover, two molecular function gene sets were identified: 'Quaternary Ammonium Group Transmembrane Transporter Activity' and 'Actinin Binding' (Fig 4). The association of 'Telomere Maintenance Via Recombination' with the biological mechanisms of COPD and frailty was a significant finding. Evidence suggests a link between

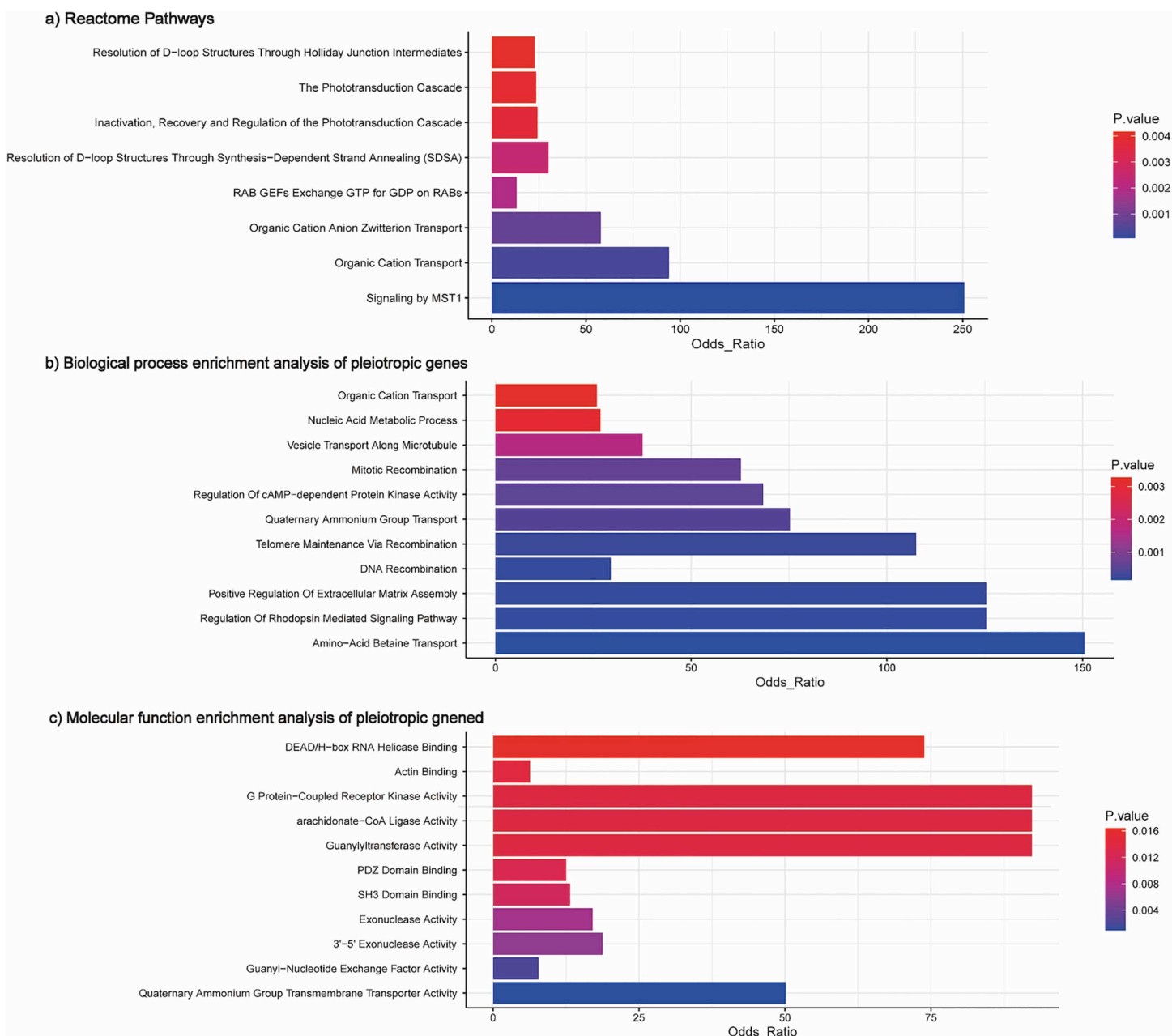

**Fig 4. Phenotype and tissue specificity for candidate pleiotropic genes.** a) Expression analysis of pleiotropic genes in GTEX tissues. b) Biological process enrichment analysis of pleiotropic genes. c) GO cellular component enrichment analysis of pleiotropic genes. d) GO cellular component enrichment analysis of pleiotropic genes.

telomere length and both frailty and COPD, emphasising the significance of telomere dynamics in these conditions [32,33] (Supplementary table 5 in S1 File).

## Transcriptome-wide association studies

We conducted a TWAS analysis of 35 GTEx tissues to assess the relationship between COPD and frailty with gene expression. Following multiple corrections, 25 gene-tissue pairs exhibited significant correlations with COPD and frailty (single-trait results are presented in

Supplementary Table 4 in S1 File). These pairs were predominantly identified on chromosomes 3, 4, 5, 6, 9, and 14.

On chromosome 4, GRK4 significantly displayed expression-trait associations in 10 different tissues, including adipose tissue and the cerebral cortex and colon. The most robust expression-trait associations for COPD and frailty were observed in fibroblast transformation and the cerebral cortex ($P_{\text{TWAS-frailty}} = 8.26 \times 10^{-5}$, $P_{\text{TWAS-COPD}} = 2.42 \times 10^{-5}$). Additionally, HTT emerged as the sole gene associated with lung tissue (the most robustly associated gene on chromosome 4, $P_{\text{TWAS-frailty}} = 1.34 \times 10^{-6}$). This aligns with findings in a published report identifying HTT as a potential COPD risk gene [34].

SLC22A5, located on chromosome 5, exhibited a significant association with COPD and frailty across multiple tissues, including adipose tissue and the oesophagus, lungs, skin, and whole blood. It displayed consistent association strengths in the lung, oesophagus, and whole blood ($P_{\text{TWAS-COPD}} = 6.23 \times 10^{-7}$).

HLA-DRB1, HLA-DQA1, and HLA-DQB1, all located on chromosome 6, exhibited significant associations with 29 tissue types, including the brain, oesophagus, stomach, and lungs. The most robust expression-trait association for frailty with HLA-DQB1 was observed in the cerebellum ($P_{\text{TWAS-frailty}} = 7.17 \times 10^{-18}$). However, for HLA-DRB1, a robust relationship with frailty was observed in the terminal ileum ($P_{\text{TWAS-frailty}} = 4.04 \times 10^{-11}$). The most prominent expression-trait association for COPD occurred at the tibial nerve ($P_{\text{TWAS-frailty}} = 3.94 \times 10^{-5}$). Functioning as pseudogenes in the MHC region, HLA-DRB1, HLA-DQA1, and HLA-DQB1 are associated with various diseases, including frailty and COPD [16,35].

Chromosome 14 exhibited the strongest expression-trait associations. KLC1 showed the highest correlation with COPD in the transverse colon ($P_{\text{TWAS-COPD}} = 9.66 \times 10^{-8}$, $P_{\text{TWAS-}}$frailty $= 4.51 \times 10^{-4}$). Furthermore, RP11-73M18.8 displayed the most significant expression-trait correlation with frailty in the cerebral cortex ($P_{\text{TWAS-frailty}} = 1.56 \times 10^{-4}$) (Supplementary table 4–6 in S1 File).

## Discussion

This study systematically explored genetic correlations between COPD and frailty using GWAS and gene-expression data, revealing a positive causal relationship and strong genetic correlation ($R_g = 0.4324$). This aligns with clinical and epidemiological evidence indicating the frequent co-occurrence of COPD and frailty [7–9]. Previous MR studies have confirmed a bidirectional association between COPD and frailty, suggesting a mediating role of comorbid genetic factors between these two diseases [36]. Through detailed analysis of genetic correlations and gene-expression patterns, our study has furthered the understanding of comorbidity mechanisms underlying COPD and frailty. Our polygenic-overlap analysis indicated that COPD and frailty share about 3.2 k variants. This genetic overlap was subjected to detailed analysis for pleiotropy and causality, as indicated by pleiotropic loci identified in ConjFDR. Shared genes revealed via TWAS were further examined, and causal relationships were verified through MR. Thus, our findings offer crucial insights into the shared genetic structure influencing COPD and frailty incidence. Additionally, we identified significant local genetic correlations in several specific genomic regions, including 3q39, 10q78, and 15q31. Genetic correlations are significant exclusively when a substantial number of variants associated with both phenotypes show a consistent direction (the same or opposite, but not mixed) [37]. All 16 sites shared between COPD and frailty that were identified by conjFDR exhibited consistent effect directions for both conditions. Consequently, the shared variants primarily associated with COPD are linked to increased frailty. These results shed light on mechanisms underlying higher frailty incidence in patients with COPD. They also suggest that interventions targeting these shared loci or their affected biological pathways, coupled with early

control of COPD or frailty progression, may offer dual effects in preventing or treating these conditions.

Functional analysis elucidated the biological significance of implicated genes that are shared between COPD and frailty, highlighting pathways that include 'Telomere Maintenance Via Recombination' and 'Organic Cation Transport' in the GO and Reactome databases. 'Telomere Maintenance Via Recombination' is pivotal in both conditions. Telomere attrition is a key mechanism in COPD pathology [33] and telomere length affects frailty onset [32,38]. Carnitine is crucial for skeletal muscle metabolism. However, its deficiency contributes to frailty, with L-carnitine supplementation showing promise in enhancing skeletal muscle function, increasing exercise tolerance, and reducing fatigue [39]. Organic cation transporters—an essential mechanism for cellular carnitine uptake—influence frailty, and genetic variants in these transporters can induce frailty by affecting carnitine deficiency [39]. Development of therapeutic drugs targeting organic cation transporters for COPD treatment is underway [40,41].

We emphasised the substantial role of the HLA region (several lead SNPs) in the interplay between COPD and frailty. This region, comprising > 200 genes on chromosome 6, is renowned for its extensive pleiotropy in various complex diseases, particularly those involving immune processes. In our TWAS analysis [37], several tissue gene pairs shared between COPD and frailty were identified, with HLA-DQA1 and HLA-DQB1 prevalent across most tissues for both conditions, suggesting their biological significance. HLA-DQB1 encodes a molecule pivotal for MHC class II antigen presentation that is linked to diverse inflammatory and autoimmune conditions, with amino acid variations in the gene product driving signalling within the MHC region [35]. HLA-DQA1 is closely associated with lung function (FEV1/FVC) [42,43], suggesting its potential role in frailty development [16]. Muscle weakness is a fundamental frailty and sarcopenia component. In a large-scale GWAS on muscle weakness in Europeans, HLA-DQA1 variations emerged as the most strongly associated among 15 loci related to muscle weakness. These genetic variants are linked to autoimmune diseases, aging, and sustained grip strength [44].

Using conjFDR to analyse GWAS results for multiple traits or conditions offers a significant advantage. It enables shared genetic signal identification that may not reach genome-wide significance in single-trait analyses by comparing and combining evidence of associations across various traits. We discovered several significant loci shared between COPD and frailty, including lead SNPs rs12441344, rs2631360, and rs10901530, shedding light on the genetic comorbidity underlying these conditions. Furthermore, organic cation transporter 2 (OCTN2), a cation/carnitine transporter, is crucial in COPD drug therapy, affecting the distribution and absorption of COPD medications in the lungs. OCT/Ns are emerging as potential treatment targets for various pulmonary conditions, including COPD [40,41]. Furthermore, carnitine, a vital molecule for skeletal muscle metabolism, is crucial in preserving muscle function and suppressing ongoing inflammation. Carnitine deficiency, linked to OCTN2 dysfunction, contributes to frailty progression while exacerbating COPD conditions. The role of OCTN2 in mediating carnitine uptake and its effect on immune functions and inflammatory responses support this connection [39]. The TGF-β/SMAD3 signalling pathway is significant in COPD pathogenesis [29], while GDF11, identified as a frailty factor, induces muscle atrophy through the SMAD2 signalling pathway [30]. Further experimental studies are needed to explore the detailed functional annotations of these lead SNPs, especially their roles in COPD and frailty development and progression.

Integrating GWAS and GTEx tissue expression data, findings from the TWAS suggest a hypothesis of a common gene-tissue mechanism for both conditions. Besides known associations with lungs, muscles, and adipose tissue, our TWAS and tissue enrichment analysis

revealed shared regulatory characteristics in the digestive system, particularly the transverse colon. This suggests that shared pathways may extend to various organs. Increasing evidence highlights the potential significance of digestive system dysfunction in inflammatory diseases, including COPD. Clinical *in vivo* and *in vitro* research confirms bidirectional gut-lung interactions in COPD, highlighting the effect of lung health on the gastrointestinal tract and the pivotal role of the gut in maintaining immune equilibrium. The "gut-lung axis" concept is gaining increasing interest [45]. Moreover, alterations in the gut microbiome can affect frailty occurrence by reducing short-chain fatty acids, decreasing ATP levels and muscle fibre metabolic efficiency, and promoting muscle mass loss [46]. Further investigation is needed to explore the precise mechanisms through which the gut influences both conditions. Developing targeted gut-focused anti-inflammatory and/or nutritional interventions is essential for complementing existing therapies and mitigating COPD or frailty progression and their associated complications.

Despite the above findings, our study has several limitations. Firstly, the use of data from populations of European descent limits the universality of our results. Future studies should include a more diverse range of ancestries. Secondly, assessing common non-genetic risks (such as environmental and social factors) for the incidence and mortality of COPD and frailty is of great importance; for instance, the effects of other comorbidities or medications on these two conditions should be explored. The current study was limited to assessing shared genetic factors between COPD and frailty, and future studies on environmental factors shared between them are needed.

## Conclusions

In conclusion, our study demonstrates the genetic correlations between COPD and frailty, offering novel insights into their shared genetic structure. This enhances comprehension of the previously reported association, suggesting potential biological mechanisms that link them. However, future GWAS with larger sample sizes are essential to comprehensively elucidate the genetic foundation of the these conditions and facilitate treatment of comorbid COPD and frailty.

## Supporting information

**S1 File. Supporting information contains supporting table 1–6.**
(XLSX)

## Author contributions

**Conceptualization:** Tong Wu, Qiang Meng, Feng Qu.

**Data curation:** Fuhui Yan, Tong Wu, Qiang Meng, Feng Qu.

**Formal analysis:** Fuhui Yan, Tong Wu.

**Methodology:** Fuhui Yan, Qiang Meng.

**Project administration:** Feng Qu.

**Resources:** Qiang Meng.

**Supervision:** Feng Qu.

**Writing – original draft:** Fuhui Yan.

**Writing – review & editing:** Fuhui Yan, Qiang Meng, Feng Qu.

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
