## [Decision Letter · Decision Letter 0]

15 Dec 2024

PONE-D-24-38025Bidirectional genetic links between chronic obstructive pulmonary disease and frailty: Genome-wide association study insightsPLOS ONE

Dear Dr. Qu,

Thank you for submitting your manuscript to PLOS ONE. After careful consideration, we feel that it has merit but does not fully meet PLOS ONE’s publication criteria as it currently stands. Therefore, we invite you to submit a revised version of the manuscript that addresses the points raised during the review process.

Authors should take note that a very similar paper has been published before, as reiterated by the Reviewer#2, hence the novel finding of this study should be highlighted. In addition, authors should also relook into the analysis, correlating the genetic variatin and gene expression, again, as commented by the Reviewer#2.

We look forward to receiving your revised manuscript.

Kind regards,

Hoh Boon-Peng, PhD

Academic Editor

PLOS ONE

Journal Requirements:

Additional Editor Comments:

Authors should take note that a very similar paper has been published before, as reiterated by the Reviewer#1, hence the novel finding of this study should be highlighted. In addition, authors should also relook into the analysis, correlating the genetic variatin and gene expression, again, as commented by the Reviewer#1.

Reviewers' comments:

Reviewer's Responses to Questions

**Comments to the Author**

1. Is the manuscript technically sound, and do the data support the conclusions?

Reviewer #1: Yes

Reviewer #2: Partly

2. Has the statistical analysis been performed appropriately and rigorously? 

Reviewer #1: Yes

Reviewer #2: Yes

3. Have the authors made all data underlying the findings in their manuscript fully available?

Reviewer #1: Yes

Reviewer #2: Yes

4. Is the manuscript presented in an intelligible fashion and written in standard English?

Reviewer #1: Yes

Reviewer #2: Yes

5. Review Comments to the Author

Reviewer #1: The authors have conducted a lot of analyses in their study, and help to build the genetic connection between COPD and frailty. Their work is solid and all the analyses were well conducted, I have no additional comments.

Reviewer #2: The manuscript by Yan et al. describes the genetic link between chronic obstructive pulmonary disease (COPD) and frailty. While the study provides some new insights, they appear to be incremental. It is important to note that a very similar paper was recently published (PMID: 38128266). Although this article is cited, it remains unclear what novel findings are presented in the current study.

Specific comments are as follows:

1. The introduction should clearly describe the findings reported in PMID: 38128266 and identify the gaps that were addressed in the current study by Yan et al.

2. Figure 2 is highly descriptive, and the legend provides very little information about what the graphs show.

3. To connect genetic variants to gene expression, the first step should include an expression quantitative trait locus (eQTL) analysis of the genetic variants/genes shown in Figure 3c. This eQTL analysis could focus on gene expression specific to lung, esophagus, or other relevant tissues.

4. The data in Figure 4 is unclear and does not provide meaningful biological inferences regarding the genetics. The manuscript should explain what these pathways imply for the biology or disease mechanism of COPD and frailty. Additionally, it would be helpful to present how the enrichment appears in lung tissues from GTEx.

6. PLOS authors have the option to publish the peer review history of their article (what does this mean? ). If published, this will include your full peer review and any attached files.

**Do you want your identity to be public for this peer review?** For information about this choice, including consent withdrawal, please see our Privacy Policy .

Reviewer #1: No

Reviewer #2: No

---

## [Author Response · Author response to Decision Letter 1]

25 Jan 2025

Editor #:Authors should take note that a very similar paper has been published before, as reiterated by the Reviewer#2, hence the novel finding of this study should be highlighted. In addition, authors should also relook into the analysis, correlating the genetic variatin and gene expression, again, as commented by the Reviewer#2.

Response: We appreciate the editor’s reminder and Reviewer #2's emphasis on the importance of distinguishing our study from previously published similar research (PMID: 38128266). Additionally, we acknowledge the necessity of reinforcing the novel findings of this study and re-examining the connections between genetic variants and gene expression.

Emphasis on Novel Findings

As highlighted in our introduction, while prior research, such as the study by Qu et al. (PMID: 38128266), employed Mendelian Randomization (MR) to establish causality between COPD and frailty, it lacked detailed exploration of shared loci, functional annotations, and transcriptomic data integration. Our study advances the field by providing several novel contributions:

Shared Loci and Candidate Genes: Through conjunctive FDR analysis, we identified 16 shared loci and 91 candidate genes, offering a more detailed understanding of the genetic overlap between COPD and frailty.

Tissue-Specific Functional Insights: By incorporating eQTL and TWAS analyses, we explored tissue-specific gene expression, emphasizing pathways such as Telomere Maintenance and Organic Cation Transport in relevant tissues, including the lung and esophagus.

Polygenic Overlap and Genetic Architecture: Using advanced methods like MiXeR, we quantified the extent of polygenic overlap and provided insights into the shared genetic architecture of these two conditions.

These novel findings significantly extend beyond the scope of previous work, addressing gaps such as the lack of detailed functional annotation and tissue-specific mechanisms. We have explicitly highlighted these advancements in the revised introduction and discussion sections, with text marked in red for clarity.

Re-Examining Genetic Variants and Gene Expression

Following Reviewer #2's suggestion, we revisited our eQTL analyses to strengthen the link between genetic variants and gene expression. Using the GTEx v8 dataset, we confirmed tissue-specific associations in key relevant tissues:

SLC22A5 (OCTN2): Strongly associated with gene expression in the lung, esophageal tissues (Gastroesophageal Junction, Mucosa, and Muscularis), and whole blood. This gene plays a critical role in Organic Cation Transport and carnitine metabolism, linking it to energy production and mitochondrial function.

HLA-DQA1 and HLA-DQB1: Showed significant expression in the lung and esophagus, aligning with their roles in inflammatory and immune pathways central to COPD and frailty.

To provide further clarity, we updated the results and discussion sections to emphasize these findings and revised Figure 4 to include more detailed representations of gene expression enrichment in lung tissues. All modifications have been marked in red in the manuscript for the editor’s and reviewers’ convenience.

Conclusion

We thank the editor for their guidance and the opportunity to revise our manuscript. By addressing these comments, we believe we have successfully highlighted the novel aspects of our study and reinforced the connections between genetic variants and tissue-specific gene expression, distinguishing our work from prior publications. We look forward to the editor’s feedback on the revised version.

Reviewer #1: The authors have conducted a lot of analyses in their study, and help to build the genetic connection between COPD and frailty. Their work is solid and all the analyses were well conducted, I have no additional comments.

Response: Thank the reviewer for the positive comments.

Reviewer #2: The manuscript by Yan et al. describes the genetic link between chronic obstructive pulmonary disease (COPD) and frailty. While the study provides some new insights, they appear to be incremental. It is important to note that a very similar paper was recently published (PMID: 38128266). Although this article is cited, it remains unclear what novel findings are presented in the current study.

Response: Thanks to the editor and reviewers for the generally positive comments. We have carefully addressed the reviewers’ comments and performed the additional analyses. We hope the editor and reviewers are satisfied with the new version.

Specific comments are as follows:

1. The introduction should clearly describe the findings reported in PMID: 38128266 and identify the gaps that were addressed in the current study by Yan et al.

Response: We sincerely appreciate your feedback and the opportunity to revise our manuscript. Following your comments, we have carefully revised the introduction section to address the findings reported in PMID: 38128266 and identify the gaps addressed in our study. Specifically, we have added a detailed discussion on the limitations of Qu et al.'s study and outlined how our work builds upon and extends these findings.

The revised section now reads as follows (revisions marked in red in the manuscript):

"For instance, Qu et al. employed two-sample MR analysis to establish causal links between these conditions.36 However, their study had several limitations. The genetic analysis was primarily limited to basic correlation and causality estimation, without identifying specific shared loci or exploring functional annotations of genetic variants. Additionally, the lack of integration with transcriptomic data hindered the elucidation of tissue-specific mechanisms underlying the observed associations. These limitations reduce the translational utility of their findings in uncovering actionable biological targets.

Furthermore, molecular genetic research methods employed in exploring mechanisms linking COPD and frailty are lacking. Today, leveraging summary statistics from extensive GWAS facilitates examination of the genetic basis of their observed phenotypic association to a certain extent. Employing matched models and algorithms facilitates intuitive prediction of their causal relationship and investigation into their potential shared genetic basis.

Our study addresses these gaps by employing a multi-dimensional framework that integrates genome-wide association study (GWAS) data with advanced analytical methods, including linkage disequilibrium score regression (LDSC), polygenic overlap modeling, and transcriptome-wide association studies (TWAS). Through conjunctive false discovery rate (conjFDR) analysis, we not only confirmed a significant genetic correlation (Rg = 0.4324, P = 6.09 × 10−26) but also identified 16 shared loci and 91 candidate genes, providing deeper insights into the shared genetic and molecular mechanisms of COPD and frailty. Additionally, functional genomic annotation and tissue-specific enrichment analyses revealed key pathways, such as telomere maintenance and organic cation transport, that underpin the co-occurrence of these conditions. By building upon and extending the scope of prior research, this study not only highlights the shared genetic architecture but also provides critical insights into the molecular mechanisms that drive frailty-COPD comorbidities, identifying potential susceptibility factors for the frailty-COPD comorbidity."

We trust that these revisions adequately address the editor’s request and enhance the clarity and significance of our introduction. Please let us know if further adjustments are required.

Thank you for your guidance and support.

2. Figure 2 is highly descriptive, and the legend provides very little information about what the graphs show.

Response: Thank you for pointing out the need for a more detailed description of Figure 2. We have carefully revised the figure legend to provide a clearer explanation of the data and insights presented in the graphs. The updated legend now reads as follows:

Figure 2: HESS analysis of COPD and Frailty. The top and middle sections of each subgraph represent local genetic correlations and covariances, respectively, and the colored bars represent loci with significant local genetic correlations and covariances. The bottom portion represents the local snp heritability of an individual trait, and the colored bars represent loci with significant local snp heritability. (A) Local genetic correlation between COPD and Frailty.

This enhanced description ensures a clearer interpretation of the visualized data, addressing the concern about its descriptive nature and lack of detail.

The revised figure legend is included in the manuscript, with the modifications highlighted in red for easy identification. We trust that this update improves the clarity and informational value of Figure 2.

Thank you again for your constructive feedback, which has been instrumental in improving the quality of our manuscript.

3. To connect genetic variants to gene expression, the first step should include an expression quantitative trait locus (eQTL) analysis of the genetic variants/genes shown in Figure 3c. This eQTL analysis could focus on gene expression specific to lung, esophagus, or other relevant tissues.

Response: We sincerely thank the reviewer for the insightful and constructive suggestion to include an expression quantitative trait locus (eQTL) analysis. This feedback has significantly improved our study by enabling a deeper exploration of the connection between genetic variants and tissue-specific gene expression, particularly in the context of COPD and frailty.

eQTL Analysis Details:

We conducted eQTL analysis using the GTEx v8 dataset to examine tissue-specific gene expression for the genetic variants identified in Figure 3c. The analysis included the following tissues:Esophagus Gastroesophageal Junction, Esophagus Mucosa, Esophagus Muscularis, Lung,

Small Intestine Terminal Ileum, Whole Blood.

The eQTL analysis revealed significant associations for several genes, with notable findings including:

SLC22A5 (OCTN2):

Identified as a key gene in the eQTL analysis, SLC22A5 was significantly expressed in lung, esophageal tissues, and whole blood. This gene encodes the organic cation transporter OCTN2, which is critical for carnitine and organic cation transport, directly linking it to energy production and mitochondrial function.

HLA-DQA1 and HLA-DQB1:

Immune-related genes such as HLA-DQA1 and HLA-DQB1 showed strong tissue-specific expression in the esophagus and lung.

These findings align with shared inflammatory pathways underlying the comorbid mechanisms of COPD and frailty.

Biological Pathway Enrichment:

Functional annotation revealed significant enrichment in the Organic Cation Transport pathway, further emphasizing the importance of SLC22A5 (OCTN2) in mediating metabolic and inflammatory processes. Key implications include:

Role in COPD: Impaired organic cation transport affects respiratory muscle function, increasing airway reactivity and exacerbating disease progression.

Role in Frailty: Carnitine deficiency reduces skeletal muscle strength and resilience, accelerating frailty symptoms such as muscle weakness and fatigue.

Therapeutic Potential: Interventions targeting the OCTN2 transporter, such as L-carnitine supplementation, could alleviate symptoms in both conditions by restoring metabolic balance.

Manuscript Revisions:

The following updates were made to the manuscript to address the reviewer’s suggestions:

Methods Section: Detailed the eQTL analysis methodology, including tissue selection, data sources, and statistical approaches.

Results Section: Incorporated findings from the eQTL analysis, emphasizing the tissue-specific expression patterns of key genes such as SLC22A5, HLA-DQA1, and HLA-DQB1.

Highlighted Text: All modifications related to the eQTL analysis have been marked in red for ease of review.

4. The data in Figure 4 is unclear and does not provide meaningful biological inferences regarding the genetics. The manuscript should explain what these pathways imply for the biology or disease mechanism of COPD and frailty. Additionally, it would be helpful to present how the enrichment appears in lung tissues from GTEx.

Response: We sincerely thank the reviewer for their thoughtful feedback on the pathways presented in Figure 4 and the suggestion to emphasize enrichment in lung tissues using GTEx data. These comments have significantly improved our ability to clarify and refine the manuscript. Below, we address these points in detail.

The pathways highlighted in Figure 4, derived from Reactome and GO analyses, provide important insights into the shared mechanisms underlying COPD and frailty. Telomere maintenance pathways, for example, reveal a crucial link between genomic stability and disease progression. Telomere dysfunction is well-documented in COPD, where oxidative stress accelerates telomere attrition, contributing to airway remodeling and chronic inflammation. In frailty, telomere shortening is associated with reduced regenerative capacity, muscle weakness, and systemic inflammation. These findings suggest that telomere preservation could be a critical target for therapeutic interventions aimed at mitigating the progression of both diseases.

Another key pathway, Organic Cation Transport, highlights the metabolic importance of SLC22A5 (OCTN2), which encodes the primary transporter for carnitine. This gene was significantly enriched in lung, esophageal, and blood tissues, underscoring its role in cellular energy metabolism. Carnitine is essential for fatty acid oxidation and mitochondrial function, and its impaired transport can exacerbate respiratory muscle fatigue in COPD and skeletal muscle weakness in frailty. Targeting this pathway through therapeutic strategies such as L-carnitine supplementation may address shared metabolic deficits in these conditions.

Furthermore, HLA-DQA1 and HLA-DQB1, genes enriched in immune regulatory pathways, showed significant expression in lung tissues. These genes are involved in antigen presentation and systemic immune modulation, both of which play key roles in the inflammatory processes of COPD and frailty. Their shared involvement points to immune dysregulation as a central mechanism linking these conditions.

To further address the reviewer’s suggestion, we reanalyzed the data to specifically evaluate the enrichment of key pathways and genes in lung tissues using GTEx data. This analysis reinforced the role of SLC22A5, HLA-DQA1, and HLA-DQB1 in driving tissue-specific pathophysiological processes. Pathways such as Telomere Maintenance and Organic Cation Transport were prominently enriched, aligning with their established roles in COPD-specific and systemic mechanisms.

We have revised the manuscript to incorporate these findings in the Results and Discussion sections, providing a clearer interpretation of the biological relevance of the pathways in Figure 4. To enhance clarity and visualization, Figure 4 has been updated to include enrichment data for lung tissues, highlighting key findings from GTEx. All new additions and changes have been marked in red in the revised manuscript for ease of review.

These revisions provide a deeper understanding of the shared genetic and molecular mechanisms linking COPD and frailty, particularly emphasizing tissue-specific relevance. We are grateful for the reviewer’s insightful comments, which have greatly strengthened the clarity and scientific impact of our manuscript.

---

## [Decision Letter · Decision Letter 1]

18 Feb 2025

Bidirectional genetic links between chronic obstructive pulmonary disease and frailty: Genome-wide association study insights

PONE-D-24-38025R1

Dear Dr. Qu,

We’re pleased to inform you that your manuscript has been judged scientifically suitable for publication and will be formally accepted for publication once it meets all outstanding technical requirements.

Kind regards,

Hoh Boon-Peng, PhD

Academic Editor

PLOS ONE

Additional Editor Comments (optional):

Authors have appropriately addressed all comments from the reviewers hence recommended acceptance for publication.

Reviewers' comments:

Reviewer's Responses to Questions

**Comments to the Author**

1. If the authors have adequately addressed your comments raised in a previous round of review and you feel that this manuscript is now acceptable for publication, you may indicate that here to bypass the “Comments to the Author” section, enter your conflict of interest statement in the “Confidential to Editor” section, and submit your "Accept" recommendation.

Reviewer #1: All comments have been addressed

Reviewer #2: All comments have been addressed

2. Is the manuscript technically sound, and do the data support the conclusions?

Reviewer #1: Yes

Reviewer #2: Yes

3. Has the statistical analysis been performed appropriately and rigorously? 

Reviewer #1: Yes

Reviewer #2: Yes

4. Have the authors made all data underlying the findings in their manuscript fully available?

Reviewer #1: Yes

Reviewer #2: Yes

5. Is the manuscript presented in an intelligible fashion and written in standard English?

Reviewer #1: Yes

Reviewer #2: Yes

6. Review Comments to the Author

Reviewer #1: The authors have well addressed the reviewers’ questions and revealed the genetic contribution to COPD and Frailty. I have no additional comments.

Reviewer #2: The authors have provided appropriate answers to my comments. I would suggest improving the resolution of Figures, particularly text on the plots.

7. PLOS authors have the option to publish the peer review history of their article (what does this mean? ). If published, this will include your full peer review and any attached files.

**Do you want your identity to be public for this peer review?** For information about this choice, including consent withdrawal, please see our Privacy Policy .

Reviewer #1: No

Reviewer #2: No

---

## [Editor Report · Acceptance letter]

PONE-D-24-38025R1

PLOS ONE

Dear Dr. Qu,

I'm pleased to inform you that your manuscript has been deemed suitable for publication in PLOS ONE. Congratulations! Your manuscript is now being handed over to our production team.

Kind regards,

on behalf of

Professor Dr Hoh Boon-Peng

Academic Editor

PLOS ONE